# Advocating for text input in multi-speaker text-to-speech systems

*Gérard Bailly[1], Martin Lenglet[1], Olivier Perrotin[1], Esther Klabbers[2]*

[1] GIPSA-Lab, GrenobleAlpes Univ., 38000 Grenoble, France
[2] ReadSpeaker, the Netherlands
`firstname.name@gipsa-lab.fr, esther.judd@readspeaker.com`

## Abstract

Nowadays text-to-speech synthesis (TTS) systems are most commonly trained using phonetic input. This is mostly due to the poor performance of the letter-to-sound (L2S) mapping (in particular with languages with opaque orthography) performed by end-to-end TTS: the empirical distribution of the words sampled in the sole training corpus cannot compete with pronunciation dictionaries. Taylor and Richmond [1] actually reported letter-to-sound errors – implicitly performed by end-to-end systems from raw text input – close to 10%.

This paper nevertheless shows that speakers produce lawful phonological variations and that end-to-end TTS systems trained to accept text input – once trained adequately – can capture these variations of pronunciation that are strong markers of sociolinguistic features. We illustrate such variations on liaisons and schwas in French and r-linking in British English. We therefore advocate for restoring text input for TTS, so that the many aspects of style variations (produced by speakers as well as stylistic variations) encoded by suprasegmental features can also be reflected in actual variations of pronunciation.

**Index Terms**: text-to-speech synthesis, multi-speaker, letter-to-sound, mixed-input

## 1. Introduction

Sociophonetics [2] studies how socially constructed variations in the sound system are used and learned. There are in fact large differences in pronunciation between regions (e.g. phonological variations of American Spanish [3]), social classes (e.g. the derhoticisation of high-class Scottish English [4]), ethnicities [5], genders, sexes, sexual orientations (e.g. realization of syllable-final /s/ as a function of sexual orientation in Puerto Rican Spanish [6]), ages [7], and within speakers.

We propose here to add a phone predictor to state-of-the art end-to-end TTS systems in order to train their text encoder to accept text input. We show that the text encoder can then be properly biased by speaker embeddings in order to generate variations of pronunciation that are strong markers of sociophonetic features.

We illustrate the generation of sociophonetic variations on liaisons and schwas in French and r-linking in British English, using two state-of-the art multi-speaker end-to-end TTS architectures: Tacotron2 [8] and FastSpeech 2 [9].

## 2. State of the art

### 2.1. Sociophonetics and phonological variations in French

Several works have studied phonological variations in French. Sources of variations are numerous: stylistic, idiosyncratic, socio-dialectal or resulting from multilingual environments or education. Brognaux et al. [10] studied 3 sociophonetic variations in French:

- *schwa deletion* in monosyllabic grammatical words (such as in "j(e) pense") and at the initial syllable of polysyllabic words (such as in "il lui a d(e)mandé")
- *liaison* i.e. the phenomenon whereby a latent final consonant in a word (Word-1) may or may not be pronounced as the onset of a following vowel-initial word (Word-2), such as in "ils vont-(t)-au cinéma"
- *deletion* of /l/ and /ʁ/ in word-final obstruent-liquid clusters, such as "in pénib(le)" or in the singular personal clitic subject pronouns, such as in "i(l) va".

They analyzed a 13-hour speech corpus, including productions of 120 speakers originating from 3 French-speaking countries (Belgium, France and Switzerland) and recorded in two different tasks (reading and conversation). They found an important effect of speaking style on schwa distribution at the start of polysyllabic words and in grammatical items, as well as on liquid deletion in word-final obstruent-liquid and in 3rd personal clitic subjects pronouns. They also found an effect of age on liaison distribution.

Adda et al. [11] studied data from read speech (BREF: 66,500 sentences from 120 speakers) vs. spontaneous speech (MASK: 38,000 sentences from 409 speakers). For both corpora, the speakers displayed no marked accent. They show that liaison realization rate for MASK is significantly lower than for BREF. Similarly, the schwa occurs more frequently in BREF than MASK.

Another aspect of sociophonetic variations is its conscious use as a social marker. Jacques Chirac, former president of France, was playing with liaisons by realizing forbidden liaisons, producing some liaisons without "enchaînement" [12] (i.e. keeping the consonant as coda of the source syllable instead of migrating to the onset of the next word: "il faut avouer" pronounced as [il fɔt avwe] instead of [il fɔ tavwe]. He performed 33.7% of liaisons without "enchaînement" in one discourse in 1981 while only .35% occurred in the PFC[1] corpus [13]. Such "prestigious liaisons" are often used as markers of social position: signaling the mastering of language.

### 2.2. TTS and phonological variations

State-of-the-art TTS systems have different ways to cope with phonological variations, either by explicitly choosing among pronunciation variants or implicitly biasing latent representations that are built along the text-to-signal mapping.

Most TTS use a pronunciation lexicon (e.g. CMUDict) where each word is assigned with a phonetic transcription that

---

[1] https://www.ortolang.fr/market/corpora/pfc

represents its canonical form, i.e. its standard pronunciation in the language the system is designed for. As an example, the phonetic input for "the" is [ðə]: it's up to further processing to harmonize the vowel before a vowel sound such as in [ðɪ ˈeɪʤ]. Such contextualization rules can be implemented by augmenting entries with latent phones, phonotactic or morpho-syntactic tags, or by post-processing rules. Note that letter-to-sound (L2S) front-ends have poor performance on lexicons: [14] report 4.6% Phoneme Error Rate (PER) vs. 19.88% Word Error Rate (WER) on the CMUDict dataset using Token-Level Ensemble Distillation, while [15] report similar performance on CMUDict, Pronlex and NetTalk using encoder-decoder models. L2S front-ends for phonetic-to-speech synthesis are thus likely to produce similar PER as implicit L2S conversion reported by end-to-end TTS for English [1] or French [16].

Moreover speakers frequently produce variants that deviate markedly from the canonical form, as underlined in the previous section. One solution is to rely on speaker and style embeddings to properly bias the output of "text" encoders fed by the canonical phonetic transcription of the input text ... already flawed by a speaker-independent L2S front-end. Typically, an L2S front-end performs text normalization to expand numbers, acronyms, etc., then looks up pronunciations of words in a pronunciation lexicon, and finally predicts pronunciations using an L2S model for words not in the lexicon. Another solution is to bias the L2S front-end by speaker- and style-specific rules of phonological variations [17] ... hoping that segmental and suprasegmental structures will match.

Our hypothesis is that feeding end-to-end TTS with text enables speaker and style embeddings to properly bias the latent phonological representations in a coherent way, including pronunciation variants, syllabification. phrasing and intonation, while keeping a high spelling accuracy. This paper is the first step towards this goal. Section 3 first introduces a new method to learn a TTS model with text input in an end-to-end manner. Section 4 then demonstrates the successful effect of speaker embedding on text input in the generation of pronunciation variants.

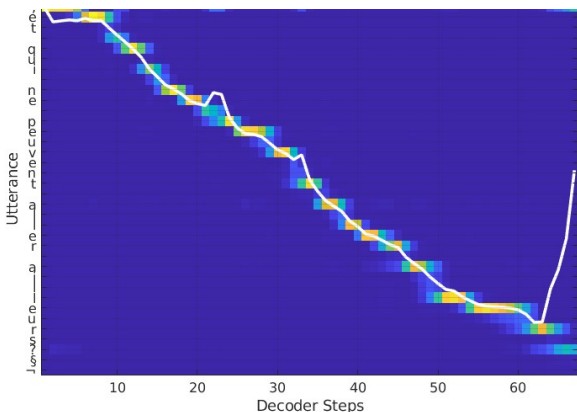

Figure 1: *Activations in the attention map for the sentence "et qui peuvent aller ailleurs?" showing silent letters (e.g. no attention weight on "u" in "qui", "en" in "peuvent" and "il" in "ailleurs") and realization of an optional liaison (attention weight on the "t" in "peuvent⏜aller"). Note that, for contextualizing the utterance, the sentence is prefixed by the punctuation(s) ending the previous one ("," here) and postfixed by the paragraph symbol ("§") and the punctuation(s) beginning the next utterance if any (speaking turn "¬" here). Final frames often pay attention to both places.*

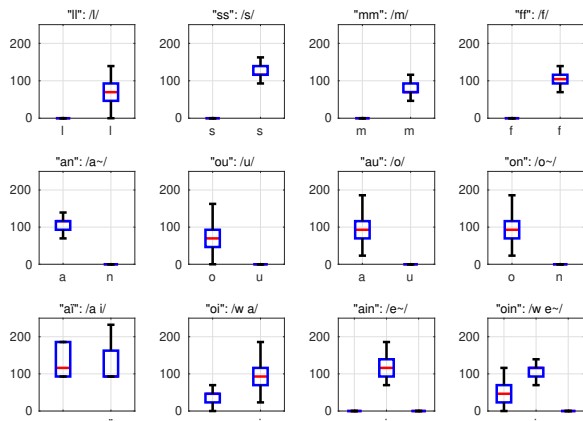

Figure 2: *Distributions of durations of activation (ms) of character sequences: when one phoneme is encoded by two letters, the second character gets mostly activated in double consonants, while the first is activated for vowels.*

# 3. Proposed framework: mixed input and phone prediction

We trained an end-to-end TTS system with both text and phonetic input, in a way similar to the representation mixing proposed by [18], and assigned two tasks to the front-end text encoder: speech and phone prediction. When a proper letter-to-sound alignment is performed prior to the mixing of text and phonetic input, end-to-end TTS are capable of remarkable performance of L2S prediction (see Figure 3).

### 3.1. Representation mixing and letter-to-sound alignment

If letter-to-sound alignment is a many-to-many mapping (a cluster of letters can correspond to one sound, and vice-versa), the text encoder of an end-to-end system is a one-to-many mapping, i.e., it assigns a sound to each individual letter input, which can be either a phone, a diphone, or in the case of letter clusters, a silence for each letter that does not carry the phone information. To train our letter-to-sound conversion system in a supervised fashion, we therefore split the task in two steps: 1) extract a set of one-to-many letter-to-sound alignment rules to create any aligned letter / sound corpus. This processed is done once and for all subsequent TTS trainings, from the analysis of a Tacotron2 attention map using non-aligned mixed input ; 2) for each new TTS model, train the phone predictor from an aligned letter / sound corpus. These two steps are implemented in Tacotron2 and described below.

#### 3.1.1. Generating a set of letter-to-sound alignment rules

Tacotron2 was first trained to predict mel-spectrograms from both text and phonetic input when available (we hand-checked the phonetic alignment of 43% of the utterances): while this representation mixing has been shown to improve spectrogram estimation [18], it also provides a letter-to-sound alignment [19] as a by-product of the Tacotron2 attention map as illustrated on Fig. 1. We can observe that for clusters of letters that produce a single sound, the attention is focused on only one letter, leaving the other ones as silent. Statistics performed on all the letter clusters of French [20] and displayed in Fig. 2 show a systematic pattern of activation for each cluster. For instance, in consonant doubling, the second letter takes the phone information

Table 1: *Multispeaker audio data used to train the French TC2.*

| Speaker | Sex | Type | #Utts | | Duration (hh:mm) | |
|---|---|---|---|---|---|---|
| | | | All | Aligned | All | Aligned |
| NEB[a] | F | Audiobooks | 81395 | 44589 | 69:45 | 34:30 |
| DG | M | Audiobooks | 20179 | 7461 | 17:16 | 6:31 |
| RO | F | Read sentences | 9940 | 584 | 10:31 | 9:57 |
| IZ | F | Scripted dialogs | 10718 | 726 | 9:17 | 0:37 |
| AD | F | Read sentences | 6506 | 6506 | 5:05 | 5:05 |
| Total | | | 128738 | 59866 | 111:54 | 56:80 |

[a] Part of this data is available at `https://zenodo.org/record/4580406`.

and the first is silent. We derived a set of rules from these statistics, that allow us to build any aligned letter / sound corpus that can be used for training phone predictors in TTS models.

### 3.1.2. Training a phone predictor plugged at the output of the text encoder

Tacotron2 was further trained to predict both mel-spectrograms and phones from all available text and phonetic input. For this, a phone predictor was adjoined to the original Tacotron2 (we will refer to this system as TC2 from now on). This prediction is simply performed by a full-connected layer with softmax. The set of target phones comprises the input phone inventory augmented with a "silent" symbol – in order to cope with silent letters, as well as spaces or mute word finals – and several "diphones" such as /k&s/, /i&j/, /d&zˆ/ (/dʒ/), etc. paired with single characters such as "x" (in "six"), "y" (in "appuyer") or "j" (as in "jazz"), respectively. We also have symbols for hiatus, syntactic vs. breath pauses, often paired with punctuations and sometimes with spaces.

Note that the text encoder and phone predictor can be trained without target audio: this enables our TC2 to be trained using aligned word dictionaries to cover the pronunciation of words not covered by the empirical distribution of the speech corpus[2]: we added L2S alignment of 95879 words from [21]. In order to cope with heterophonic homographs, we furthermore added 8137 sentences comprising at least one heterophonic homograph in context [22].

### 3.2. Data, multi-speaker embeddings and dual-task training

We trained TC2 with speech data from 5 speakers (see Table 1): four female speakers (NEB, RO, IZ and AD) and one male speaker (DG). Each corpus is first chunked into utterances by detecting silences $> 400ms$. Each utterance is then aligned with text. Note that for contextualizing utterances produced in paragraphs, each sentence is prefixed by the punctuation(s) ending the previous one ("," is added by default) and postfixed by the paragraph symbol ("§") and the punctuation(s) beginning the next utterance if any (see example in Fig. 1). Indeed, end-of-paragraphs are more likely to be associated with boundary tones than end-of-sentences inside a paragraph, e.g. often full stops inside paragraphs are sometimes marked by transitional prosody [23]. For sentences spelled in isolation, they are prefixed by "." and suffixed with "§".

Almost half of the utterances of all corpora have been

aligned with their phonetic transcriptions. All phonetic alignments have been hand-checked. Speaker-embeddings are added to all contextualized symbol embeddings computed by the text encoder and learned together with the phone predictor that is plugged after this summation.

Note that L2S alignments from dictionaries and homographs are considered as speaker-independent and thus duplicated for each speaker embedding. No attempt has been made here to simulate phonological variations. If any, phonological variations are solely provided by ground-truth speech data.

### 3.3. Training policy

All models have been trained for 100 epochs with the Adam optimizer and results are given using 10-fold cross-validation: each set of data (signals aligned with text or phones, text aligned with phones) for each speaker has been randomly partitioned into 10 parts and concatenated to build the training and test folds. Three losses are minimized for TC2: a spectrogram loss, a gate loss [24] and a phone predictor loss.

### 3.4. Phone prediction performance

We report here results of phone prediction from all available training data: 15,051,582 input letters and 1,920,108 input phones. We have 132 input symbols: 36 phones, lower and uppercase accented and non accented letters, space, punctuation and quotation marks, special symbols for emphasis, end-of-paragraph and begin/end of utterances. Note that all numbers, acronyms, etc. are spelled in full before being processed. As output and as mentioned earlier, the set of 66 target phones supplement the input phone inventory with one "silent" phone, 28 diphones and one "silence" phone (for now, we do not distinguish between breath, syntactic pauses and hiatus) to align with text input. Diphones comprise most consonants adjoined with a schwa: like in English where a schwa sound is often pronounced at the end of words with a final unstressed syllable (e.g. "present"), we cope with their eventual alignment with single letters such as "Groënland" aligned with /g r o eˆ n l a˜ _ d&q/.

Figure 3 displays the confusion matrices for (left) the prediction of L2S alignments (i.e. text input) and (right) the prediction of target phones from input phones (since mixed input is possible). L2S is quite accurate: the overall F-score is close to .99 when all 15,051,582 letters are considered while it raises to .999 when non-words symbols are discarded. The 1,920,108 input phones are correctly transcribed and only 7 minor "errors" are detected: mainly mid-open vs. mid-closed vowels, full closed vowels vs. semi-vowels. When exploring L2S prediction "errors", Tacotron2 is often right, except for loan words.

---

[2]Contemporary words such as "super" or "technique" are not mentioned in Librivox audiobooks from the beginning of the 20th century! These two examples are likely to be wrongly pronounced /sype/ and /tɛʃnik/.

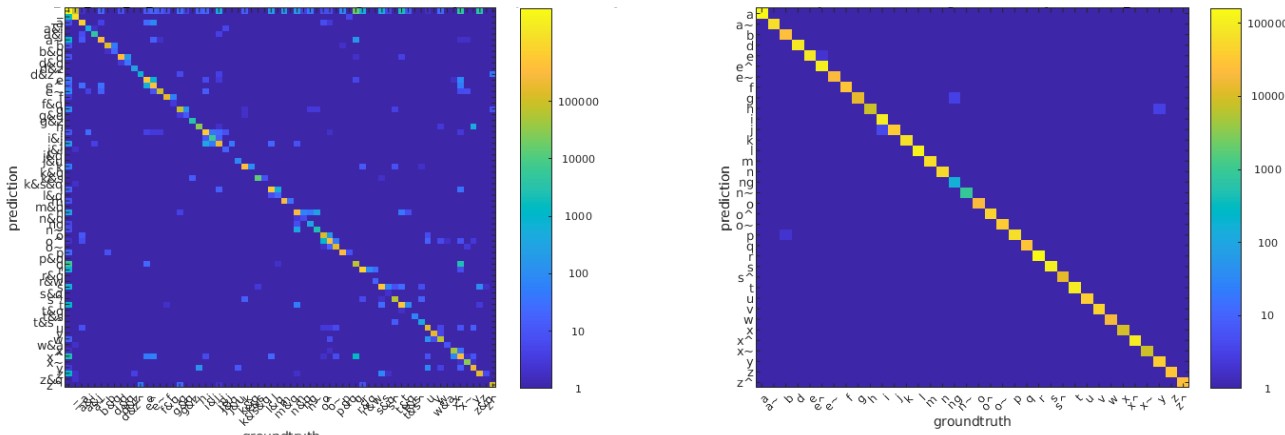

Figure 3: *Phone prediction from text vs. phonetic input. Left: the confusion matrix (displayed with log counts) features the predicted phones (ordinate) with corresponding hand-checked ground-truth alignments with text (abscissa): the F-score is close to .99 when all 15 millions characters are considered while it raises to .999 when non-words symbols are discarded. Right: the confusion matrix features the predicted vs. input phones; phonetization of punctuations are ignored here (no silent nor silence output phones); this confusion matrix is almost diagonal: we got only 9 minor "errors" among 1,920,108 phones.*

## 4. Exploring phonological variations

Using aligned data from the 3 most represented speakers (NEB, DG, AD), we will further show that speaker-specific phonological variations are implicitly captured by the speaker's bias added at the output of the text encoder before phone estimation.

### 4.1. Data and speaker embeddings

In our aligned ground-truth data for these speakers, we selected possible placements of 4 liaisons and schwas using regular expressions:

- words ending with "r" ("alle[r]‿ici"), "s"("me[s]‿amis"), "n" ("mo[n]‿oncle"), "t|d" ("ayan[t]‿été"), followed by a word beginning with a vowel or an "h" and respectively spelled as /r/, /z/, /n/, /t/ or silent
- word-internal "e" surrounded by consonants or ending a word ("rapp[e]ler") spelled as a schwa or a full mid-open vowel /œ/

We then compare the predictions of our multi-speaker TC2 model on three distinct test datasets uttered by our three speakers, respectively, (a) using for each the speaker embedding of the speaker that recorded the test dataset (this is the "Multi" embedding in the following) or (b) imposing the embedding of one speaker to the three datasets.

In condition (b), all phones of the three datasets are predicted with each speaker embedding. In that case, for each individual phone, when all predictors agree, we name these occurrences "Consensus": they are likely to spot mandatory vs. prohibited liaisons/schwas. Non-consensual occurrences are likely to spot optional pronunciations and illustrate idiosyncratic phonological variations across speakers. Note that in case of consensus, if all predictors agree between them, it can be incongruent with any speaker ground truth value. Therefore, F-scores are systematically reported for the predictions.

### 4.2. Results

Table 2 gives the overall counts and percentages of pronounced phones for our 3 speakers as well as the F-scores for consensual vs. non-consensual predictions.

#### 4.2.1. Ground truth realisation of liaisons and schwas

Note first that percentages of pronounced segments differ between speakers, since corpora have different textual contents: consensual percentages certainly reflect corpus biases. Non-consensual percentages should thus be compared to percentages of consensual percentages for each speaker.

The percentage of consensual predictions differ between the types of liaisons: while the "n" liaison is almost always consensual (only 14 are non consensual, 10 on the NEB test set and 4 on the AD test set), more phonological variation is observed on other segments. Non-consensual predictions clearly reflect idiosyncratic phonological variations: while NEB tend to over-realize all optional liaisons, DG tend to not pronounce optional "r" liaison and schwas, and AD tend to not pronounce optional liaisons at all.

#### 4.2.2. System prediction of liaisons and schwas

When all predictors across speaker embedding agree (consensus), predictions are quite effective: F-scores are close or above .9. The "n" liaison is notably predicted with a F-score of .98.

In Non Consensus, and when a speaker embedding is applied to the full dataset (last row of Table 2), we see that speaker embeddings consistently bias the pronunciation of optional liaisons and schwas: when applying the NEB bias on all data, we see that all optional liaisons "r", "s" and "t" are generated while, on the contrary, the AD bias prohibits the generation of all optional liaisons. The behaviours of predictions given each speaker embedding are thus representative of the ground truth realisation of liaison and schwas for each speaker. The F-scores of the "Multi" policy are always superior to the imposition of the embedding of a particular speaker.

## 5. Replicating the experiment with FastSpeech 2 and British English.

We ran a parallel experiment for British English with two speakers (one male RSM and one female RSF) using proprietary TTS databases. We chose to use the Blizzard2023_TTS version

Table 2: *Realisation of liaisons and schwas. For the ground-truth (Grd), counts and percentages of pronounced segments are given for the different speakers. For predictions (Prd), we provide F-scores and percentages of pronounced phones on all data for the different models. Percentages over .8 or under .2 are highlighted in blue and red, respectively.*

| Set | | Data | Embeddings | "r" → [ʁ] | "s" → [z] | "n" → [n] | "t\|d" → [t] | "e" → [ə\|œ] |
|---|---|---|---|---|---|---|---|---|
| Consensus | Grd (#/%) | NEB | / | 3295/.89 | 10068/.78 | 4522/.77 | 10258/.69 | 119086/.55 |
| | | DG | / | 562/.80 | 2151/.61 | 1014/.78 | 2596/.67 | 22707/.46 |
| | | AD | / | 643/.55 | 2793/.47 | 1214/.54 | 2567/.41 | 27331/.43 |
| | Prd (Fscore/%) | All | Multi | .89/.83 | .90/.69 | .98/.73 | .92/.64 | .90/.52 |
| Non Consensus | Grd (#/%) | NEB | / | 454/.74 | 1057/.85 | 10/.10 | 1116/.88 | 6623/.75 |
| | | DG | / | 118/.08 | 218/.32 | - | 298/.64 | 1222/.41 |
| | | AD | / | 89/.01 | 301/.18 | 4/.50 | 217/.15 | 1311/.62 |
| | Prd (Fscore/%) | All | Multi | .82/.68 | .79/.70 | .28/.71 | .73/.81 | .64/.84 |
| | | | NEB | -/.99 | -/.99 | -/.99 | -/.99 | .63/.99 |
| | | | DG | .47/.01 | .54/.23 | .28/.71 | .54/.69 | .60/.09 |
| | | | AD | -/.00 | .56/.02 | -/.00 | -/.00 | .49/.80 |

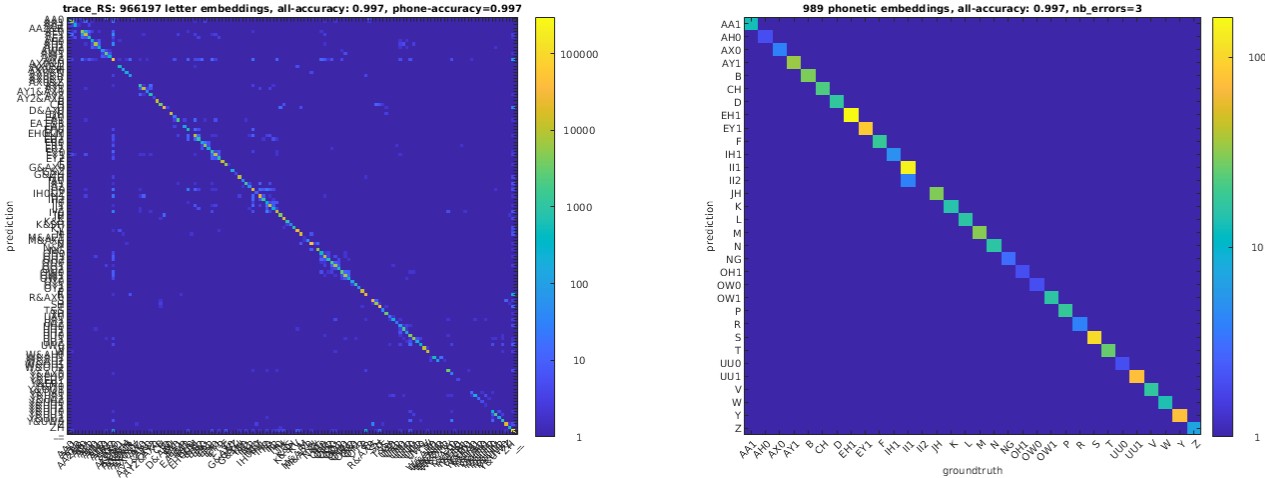

Figure 4: *Phone prediction from text vs. phonetic input for the English data. Same confusion matrices as 3. Here "only" 966197 input characters and 989 input phones (mainly acronyms) have been transcribed. F-score of L2S (.997) is quite high despite the larger set of phonetic labels compared to French (3 levels of accentuation for vowels and many diphones).*

of FastSpeech 2[3] in combination with the universal HiFiGAN vocoder. In British English, there is a phenomenon called *r-linking*. It occurs when a word ends in the letter r (which only occurs when it is preceded by a vowel) which in British English is not pronounced unless the next word starts with a vowel. In a rare number of cases, speakers may produce an *intrusive r*. This can occur when a word ends in an open vowel and the following word starts with a vowel, such as in the phrase 'law(r)-and-order". Informal analysis of the two TTS databases used for this study showed that there is a difference between the two speakers in how often they use r-linking. In cases where they don't use it, they will introduce a glottal stop to separate the two words.

For each speaker, we used 8260 sentences of data. The alignments between input characters and phones was created automatically by using *m2m-aligner* [25]. Similar to the French Tacotron 2 experiment, we ran a 10-fold cross-validation, but for 150 epochs. The phone set is similar to the CMU phone set and encodes the phones as well as the stress in vowels. After synthesizing the 10 test sets we found the sentences with phone input had perfect phone prediction as expected. The sentences with text input, which contained 949413 phones, had an

F1 score of 0.981. The automatic m2m-aligner caused some strange alignments and the model was not trained with non-audio data from our pronunciation dictionaries, which can explain some of the discrepancies. Additionally, some acronyms had not been properly aligned to the phonemes. Nevertheless, we can say that this method does work well with FastSpeech 2 and with British English data.

Table 3 shows the results for the 4066 cases where r-linking could occur. In the text we searched for words ending in "r" or "re" followed by words starting with a vowel. Speaker RSF shows a tendency to use r-linking less frequently than speaker RSM (40% vs. 65% of the time), and the same pattern is observed in the predicted phonemes when using the correct speaker embedding in the multi-speaker model. However, there were only 110 out of 4066 cases where the different models didn't agree on the predicted phone. When synthesizing biased with speaker RSM all 110 non-consensus instances were predicted with ʁ whereas for speaker RSF they were all predicted without. These experiments need to be replicated with other languages that have different patterns of phonological variations.

[3]https://github.com/MartinLenglet/Blizzard2023_TTS

Table 3: *Realisation of r-linking in British English for the two speakers RSM and RSF, giving counts of realised and unrealised r-linking (same information as Table 2).*

| Set | | Data | Embs | "r" → [ʁ] |
|---|---|---|---|---|
| Cons. | Grd (#/%) | RSM | / | 1866/.65 |
| | | RSF | / | 2090/.40 |
| | Prd (Fscore/%) | All | Multi | .80/.52 |
| Non Cons. | Grd (#/%) | RSM | / | 61/.51 |
| | | RSF | / | 49/.36 |
| | Prd (Fscore/%) | All | Multi | .52/.68 |
| | | | RSM | 1/.43 |
| | | | RSF | 0/.57 |

## 6. Conclusions

We show that text encoders of current end-to-end TTS are capable of performing quite impressive L2S mapping, given proper L2S alignment and mixed input training. We also demonstrate that feeding such systems with text instead of pre-processed phonetic input enables the systems to deal with phonological variation, in particular speaker-specific policy for generating optional phones such as liaisons or schwas.

Several phonological variations are idiosyncratic such as metathesis ([aʁeɔpɔʁ] for "aéroport" in French) or sound shifts ([bɑks] for "box" in American English). Comparing the ability of L2S front-ends vs. end-to-end TTS to capture such phonological variations in coherence with other aspects of speakers's speech is both a scientific and technological challenge.

We study here the local impact of speaker biases. But their impact surely goes beyond these elementary decisions: contextual by-effects such as the generation of pauses or contextual assimilation have to be studied. More generally, we are currently exploring the impact of speaker, emotion, or style biases on the phonological variation and how they combine.

In a nutshell, text is surely the most efficient way to specify what has to be said while biasing the output of the text encoder – or any other latent representation built as a product of the text-to-sound mapping – is the most elegant way to specify how to say it.

## 7. Acknowledgments

Part of this work is supported by BPI-France (project THERA-DIA) and ANR 19-P3IA-0003 MIAI. It was also partly performed using HPC/AI resources from GENCI-IDRIS (Grant AD011011542).

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
