# OpenReview forum: "Advocating for text input in multi-speaker text-to-speech systems"
_Interspeech.org/2023/Workshop/SSW — SSW12_

### Official Review · Reviewer_moGg · 2023-06-05
**Interesting position and discussion**

**Rating:** 6
**Confidence:** 4

**Review:**

This paper positions that the use of text (an even more end-to-end) representation can reveal ever more fully than letter-to-sound (L2S) or grapheme-to-phoneme (G2P) systems given enough data, reasonable alignment.  Moreover, the paper suggests that this direction should be pursued specifically to model those speech variants where prescriptive L2S models are incorrect -- including non-native speech,  dialects, and specifically, sociophonetic variation.

The position is, for the most part, well justified and an interesting counter argument to the position that holds that end-to-end models are best suited to capturing the "head" of the linguistic distribution and suffer when it comes to modeling "interesting" less represented phenomena, of the sort described here.

The constructive proposal is to "add a phone predictor to state-of-the-art end-to-end TTS systems in order to train their text encoder to accept text input".  Isn't this, to some extent, simply a retraining of existing L2S rules to better specific phenomena, rather than moving to the fully e2e paradigm and letting the data and model learn appropriate L2S mappings under appropriate sociophonetic conditions?

One limitation of this work is the empirical evaluation of this work is fairly narrow, focusing on a few well understood phenomena.  This leaves and open question if there is some sociolinguisic/sociophonetic phenomena that is not as well documented and understood, can this technique capture it, or reveal it, by virtue of training on text-input rather than using a L2S component?

In Section 3.4 it is implied that the input text is already normalized to some degree.  In Section 3.1.2, the model needs to predict target phones to operate.  To what extent do these components make the modeling more complicated or at least data-hungry, despite eliminating the L2S component of the modeling.  Specifically, the need for phonetic transcription of less-common phenomena is more challenging than re-using a L2S module.


Typo:
On page 2, Sections 2.2 and 3 letter-to-sound is abbreviated LTS, but in the abstract L2S is used.
In Section 5, there is a missing space in "vs65%"

---

> ### Author Response · Authors · 2023-06-16
> **retraining of existing L2S rules to better specific phenomena, vs. moving to the fully e2e paradigm**
>
> Thank you for the interesting comments in your review.
> This work put a coin in the use of an external L2S component, while integrating more and more AI components in the models (Bert, style encoders, etc). Either:
> (1) the external L2S module is trained using effective transcriptions and can be biased at some level by speaker embeddings... and then why not consider our solution that plugs and trains a phonetic predictor… in a way similar to prosodic predictors for F0, energy in FS2: we somehow force the latent space before the acoustic decoder to perform such predictions because we know it's important. And we are working on a journal paper that shows it's actually effective and improve speech quality and controlability
> (2) the external L2S module is still rule-based or uses decision trees, and it's rather difficult to get it consider such phonological variations

---

### Official Review · Reviewer_mwpx · 2023-06-06

**Rating:** 8
**Confidence:** 3

**Review:**

The paper argues for text instead of phoneme sequences as input into neural end-to-end speech synthesizers. The paper shows that the implicit G2P conversion in such a system can learn speaker-specific characteristics (such as sociophonetic and other variability) for different speakers.
I am very much convinced that including G2P into an *end-to-end* system is very very plausible and the arguments of the paper are sound.
The (little?) draw backs of this paper are two-fold: there is no evaluation of the TTS results (naturalness, ...) as compared to a baseline approach. This is aggravated by the fact that G2P as part of end-to-end TTS doesn't work well (as is openly discussed already in the abstract). Therefore, while it may be more *plausible* to include G2P into the NN model, it may still not be the correct choice for technical systems. The paper fails to quantify the cost of modeling speaker-specifics in terms of e.g. MOS.

All in all, I like the paper as I see it raising an important discussion point.

---

> ### Author Response · Authors · 2023-06-16
> **Performance of phonetic predictor**
>
> Dear reviewer,
>   The L2S alignment of training data can be improved, but we demonstrated that F-scores of phonetic predictions are rather high (.96 for our French data and .99 for the English data). Much higher that was reported in the literature so far (but using speech recognition). The remaining problems are not so easy to solve (eg. homographs), in particular rare or loan words that can be spelled using native or target approximations... and this is also sociophonetics : it shows the mastery of the language and the comprehension of the spelling context by the speaker. "gentleman" or "gentelman", English words loaned from Latin/French (ending with -tion, -sion)...
>   Demonstrating the impact of the phonetic preditor on acoustics was out of scope of this paper. But we will publish soon. Stay on-line!

---

### Decision · Program_Chairs · 2023-06-14

**Decision:**

Accept

**Comment:**

Dear authors,

SSW2003 received 45 papers. The acceptance rate is 82%. We are pleased to inform you that your paper has been ACCEPTED by the SSW2023 Programme Committee.

Please read the reviews carefully and submit your camera-ready paper by June 28th. Most of reviewers performed a detailed review. Please answer to their questions and take into account their comments. Note that camera-ready papers are credited of one extra page to allow authors to consider reviewers’ suggestions. So max 7 pages in total including figures & refs.

Regards,
The SSW organizing chairs